# Internal neural states influence the short-term effect of monocular deprivation in human adults

Yiya Chen[1,2†], Yige Gao[3,4†], Zhifen He[1,2†], Zhouyuan Sun[3,4], Yu Mao[1,2], Robert F Hess[5*], Peng Zhang[3,4,6*], Jiawei Zhou[1,2*]

[1]State Key Laboratory of Ophthalmology, Optometry and Visual Science, Wenzhou Medical University, Wenzhou, China; [2]National Engineering Research Center of Ophthalmology and Optometry, Wenzhou Medical University, Wenzhou, China; [3]State Key Laboratory of Brain and Cognitive Science, Institute of Biophysics, Chinese Academy of Sciences, Beijing, China; [4]University of Chinese Academy of Sciences, Beijing, China; [5]Department of Ophthalmology and Visual Sciences, McGill University, Montreal, Canada; [6]Institute of Artificial Intelligence, Hefei Comprehensive National Science Center, Hefei, China

**Abstract** The adult human visual system maintains the ability to be altered by sensory deprivation. What has not been considered is whether the internal neural states modulate visual sensitivity to short-term monocular deprivation. In this study we manipulated the internal neural state and reported changes in intrinsic neural oscillations with a patched eye open or closed. We investigated the influence of eye open/eye closure on the unpatched eye's contrast sensitivity and ocular dominance (OD) shifts induced by short-term monocular deprivation. The results demonstrate that internal neural states influence not only baseline contrast sensitivity but also the extent to which the adult visual system can undergo changes in ocular dominance.

*For correspondence:
robert.hess@mcgill.ca (RFH);
zhangpeng@ibp.ac.cn (PZ);
zhoujw@mail.eye.ac.cn (JZ)

[†]These authors contributed equally to this work

## Editor's evaluation

The authors report the results of three experiments assessing how one or both eyes open under a patch influence resting EEG activity, contrast sensitivity, and binocular balance in normally sighted subjects. Their results suggest that the state of eye-opening temporarily, but significantly, influences shifts in ocular dominance with relevance for the treatment of binocular visual disorders such as amblyopia that are treated with periodic monocular occlusion. The evidence supporting their conclusions is solid and the findings are important for the field.

## Introduction

Recent studies have shown that when one eye is deprived of its input for a short period of time (30 min to 2.5 hr), visual brain mechanisms undergo a neural change that results in not only a change in visual sensitivity but also interocular balance (i.e., ocular dominance) (*Lunghi et al., 2011*; *Zhou et al., 2013a*). However, unlike the long-term neuroplastic changes by abnormal visual experience in critical period, such change peaks immediately after patch removal and remains for up to 30–90 min (*Lunghi et al., 2011*; *Zhou et al., 2013a*), which we call as short-term monocular deprivation effect. A number of results suggest that this short-term monocular deprivation effect mainly involves V1 (the primary visual cortex), these include intrinsic optical imaging in monkeys (*Begum and Tso, 2015*), and magnetoencephalography (MEG) studies (*Chadnova et al., 2017*), electroencephalogram (EEG)

studies (*Lunghi et al., 2015a*; *Zhou et al., 2015*), functional magnetic resonance imaging (fMRI) studies (*Binda et al., 2018*), magnetic resonance spectroscopy (MRS) studies (*Lunghi et al., 2015b*) and psychophysics studies (*Zhou et al., 2014*) in human adults. The role played by a number of exogenous factors have been investigated, namely image properties (*Zhou et al., 2014*; *Zhou et al., 2017*), exercise (*Lunghi and Sale, 2015c*), visual pathology (*Lunghi et al., 2019b*), body mass index (*Lunghi et al., 2019a*). What is not known is whether there is a role played by internal neural states in modulating this short-term monocular deprivation effect.

It is well known that there are characteristic differences in resting-state brain activity in the absence of visual stimulation, for example, when the two eyes are open vs. closed in the dark, there is a significant decrease in occipital alpha oscillations called the Berger effect. There are power and coherence changes of a broad spectrum in the Δ, θ, $α_1$, $α_2$, $β_1$, $β_2$, and γ frequency bands that are presumed to be correlates of the switching of involuntary preliminary attention from internally directed attention specific for the eyes closed state to externally directed attention specific for the eyes open state (*Boytsova and Danko, 2010*). FMRI studies have shown that eyes open rest conditions are associated with larger activation of the visual cortex but smaller activation of the lateral geniculate nucleus (*Marx et al., 2004*). In the eyes closed state, activations of the ocular motor-related brain areas are larger, including the prefrontal eye fields, parietal and frontal eye fields, cerebellar vermis, thalamus, and basal ganglia (*Marx et al., 2004*). It has been argued that eye closure can alter the processing mode of the sensory system by decoupling geniculostriate processing in favor of enhanced thalamocortical coupling in non-visual brain areas (*Brodoehl et al., 2015*). While it remains an important question whether this type of intrinsic regulation can modulate visual sensitivity, it is a difficult question to answer because any comparison between eyes open and eyes closed conditions necessarily excludes the ability to measure visual sensitivity using external visual inputs. Using the short-term monocular deprivation paradigm described above where one eye is deprived of its visual input for a short period of time, we have been able to assess the role of the internal state (comparing eyes open with eyes close behind the deprivation patch) for visual sensitivity and changes in binocular balance in adult humans. In particular, we directly compare the monocular deprivation effects, as assessed by changes in EEG power, steady-state visually evoked potentials (SSVEPs), and contrast sensitivity, during the period when the patched eye (PE) is kept either open or closed *behind* the patch. We derive contrast gain and ocular dominance changes that occur as a result of patching one eye for a 2.5 hr period when the eye *behind* the patch is either open or closed. The results show that the short-term monocular deprivation effect can be modulated by the internal state in the absence of visual stimulation and is greater when the eye behind the occlude is kept open (i.e., eye-open state).

## Results

### The immediate effects of open vs. close of the PE on intrinsic neural oscillations and the unpatched eye's sensitivity

To investigate whether there is an internal state difference in intrinsic neural oscillations when only one eye is open or closed, we recruited 20 normal adults and measured the amplitudes of their alpha oscillations at the resting state with the two eyes closed, two eyes open, monocular patching with the PE open, and monocular patching with the PE closed. Here, we focused on the alpha oscillation because it is the strongest intrinsic neural oscillation in the brain, which is the hallmark of internal state changes (*Boytsova and Danko, 2010*). *Figure 1a* shows the amplitude spectrum averaged across subjects for the four conditions. *Figure 1b* shows the average amplitude at the alpha peak frequency. Similar to previous reports (*Boytsova and Danko, 2010*), alpha amplitude was significantly lower when two eyes are open compared to closed (t(19) = –2.272, p=0.035, two-tailed paired samples t-test with Holm-Bonferroni correction (*Holm, 1979*), number of comparisons k=2). Importantly, in the monocular patching conditions, the alpha amplitude was significantly lower when the PE remaining open as compared to the PE being kept closed (t(19) = –3.944, p<0.001, with Holm-Bonferroni correction, k=2). There were also weak signals in the beta band (*Figure 1a*, 20 Hz), but without significant changes in the eye-open than eye-closed condition.

We further measured SSVEP to a sinewave plaid at 24% contrast counterphase flickering at 7.5 Hz, and the contrast sensitivity of the unpatched eye (UPE) for both PE-open and -closed conditions. The results in *Figure 1c* show stronger SSVEP (t(19) = 2.737, p=0.026, with Holm-Bonferroni correction,

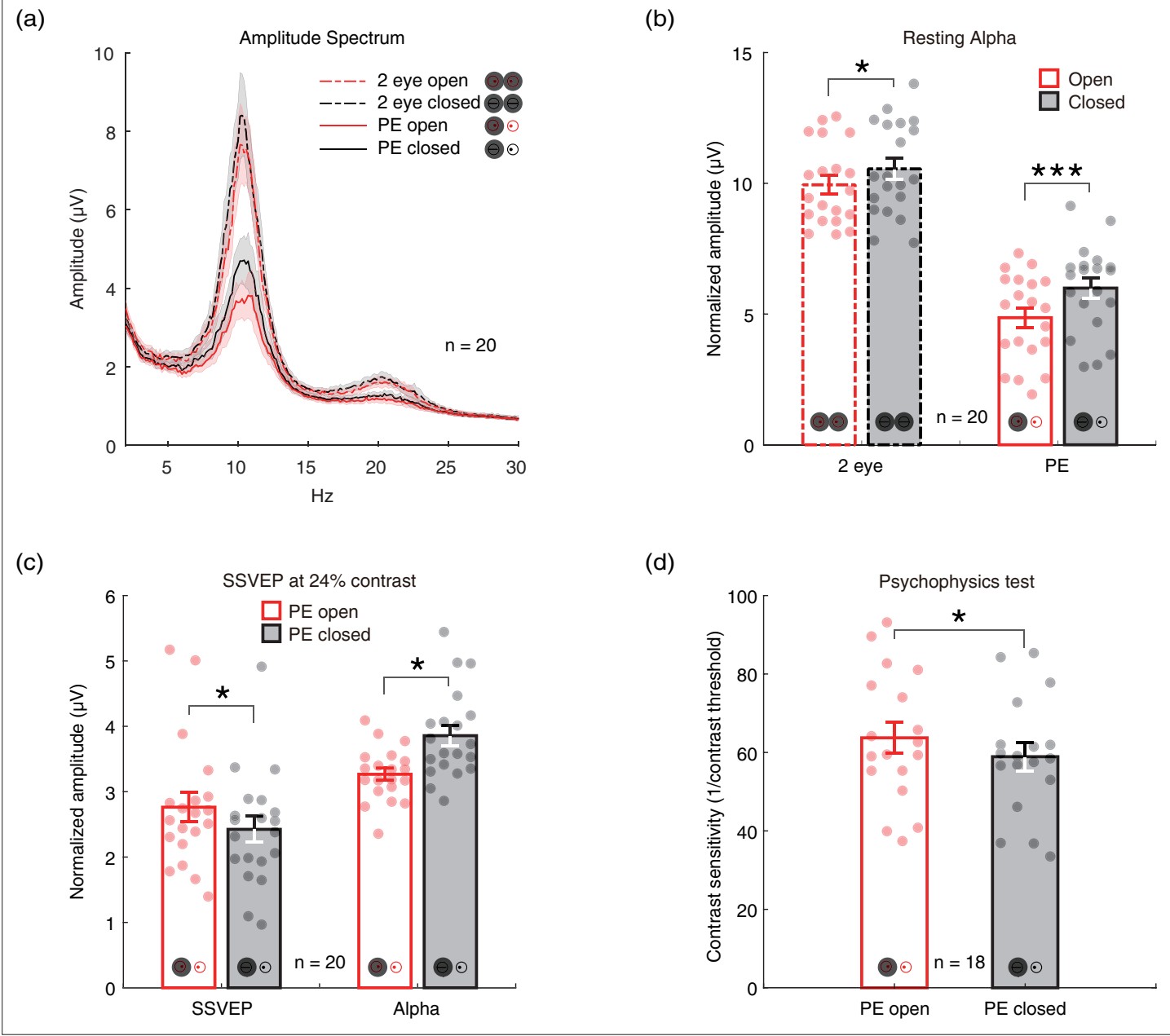

**Figure 1.** The immediate effects of open vs. close of the patched eye (PE). (**a**) The amplitude spectrum averaged across subjects between 2 and 30 Hz for four conditions (two eyes open: red dashed line; two eyes closed: black dashed lines; PE open: red solid line; PE closed: black solid line). Shaded area indicates one standard error of the mean (SEM) across 20 subjects. (**b**) The normalized amplitude at the alpha peak frequency. Each dot represents one subject. Error bars represent one SEM across 20 subjects. (**c**) The average amplitude of steady-state visually evoked potential (SSVEP) and peak alpha. Each dot represents each observer. Error bars represent one SEM across 20 subjects. (**d**) The contrast sensitivity of the unpatched eye (UPE). Each dot represents one observer. Error bars represent one SEM across 18 subjects. The black star indicates the significance of the two-tailed paired samples t-test with Holm-Bonferroni correction, *: $p < 0.05$; ***: $p < 0.001$.

The online version of this article includes the following source data for figure 1:

**Source data 1.** Related to *Figure 1a*.

**Source data 2.** Related to *Figure 1b*.

**Source data 3.** Related to *Figure 1c*.

**Source data 4.** Related to *Figure 1d*.

k=2) and weaker alpha oscillations (t(19) = –2.683, p=0.015, with Holm-Bonferroni correction, k=2) for the PE-open condition compared with the PE-closed condition. As shown by *Figure 1d*, the contrast sensitivity of the UPE with PE open was also higher than that with PE closed (t(17) = 2.667, p=0.016).

We calculated the difference between the eye-open and eye-closed conditions as: (closed-open)/open, for all five measurements: contrast sensitivity, SSVEP amplitude, peak alpha amplitude in the SSVEP sessions ($\alpha_{SSVEP}$), peak alpha amplitude in the resting state when both eyes open/closed ($\alpha_{Bi}$), and PE open/closed ($\alpha_{mono}$). Compared to the PE condition, the contrast sensitivity and SSVEP amplitude of the UPE were immediately decreased by 7.5% and 12.3%, whereas the amplitudes of $\alpha_{SSVEP}$, $\alpha_{Bi}$, and $\alpha_{mono}$ were immediately enhanced by 18.1%, 6.2%, and 23.3%, respectively, in the PE-closed condition. As shown in *Figure 2*, a positive correlation of the effect of eye open vs. eye close was found between contrast sensitivity and SSVEP amplitude (r=0.665, p=0.013), and between peak alpha amplitudes in the SSVEP sessions and resting state ($\alpha_{SSVEP}$ vs. $\alpha_{Bi}$: r=0.558, p=0.011; $\alpha_{SSVEP}$ vs. $\alpha_{mono}$: r=0.643, p=0.002; $\alpha_{Bi}$ vs. $\alpha_{mono}$: r=0.468, p=0.037). No significant correlation was found between other pairs of measurements (all p>0.05).

## The aftereffects of 2.5-hr monocular patching on contrast sensitivity

So far, we have shown that the alpha power is stronger if the eye *behind* a monocular patch is closed than when the eye *behind* the patch is open, which occurred in both the resting state and the SSVEP sessions. The effects of the PE-closed condition were to inhibit the SSVEP power and the sensitivity of the UPE during patching.

One interesting question is whether such inhibition would influence the contrast gain changes of each eye after a short-term monocular deprivation (i.e., the aftereffects of 2.5 hr monocular patching)? We measured monocular contrast sensitivity as the inverse of contrast threshold. After 2.5 hr of monocular patching in the PE-open condition, the contrast sensitivity of the UPE changed from 103.086±6.961 (mean ± SE) before patching to 70.017±6.236 immediately after removal of the patch. This means that the short-term monocular deprivation temporally decreased the contrast sensitivity of the UPE by 32.1%. In contrast, the contrast sensitivity of the UPE decreased from 90.324±6.791 to 69.324±4.685 after 2.5 hr of monocular patching in the PE-closed condition. This implies that the aftereffect of 2.5 hr of monocular patching on monocular contrast sensitivity was 34.5% less in the PE-closed condition than that in the PE-open condition.

Then, to normalize the change of contrast sensitivity for each eye at different conditions, we calculated the individual change of monocular contrast sensitivity in decibels (dB), where dB = 20 × $\log_{10}$(contrast sensitivity at post-measure session/contrast sensitivity at baseline). In *Figure 3*, we plot the average change of monocular contrast sensitivity after 2.5 hr of monocular patching where the PE remains open (sections M1&M3) and where the PE is kept closed (sections M2&M4) as open red triangle symbols and filled black triangle symbols (UPE: dashed line with inverted triangle; PE: solid line with regular triangle), respectively. The contrast sensitivity of the UPE becomes smaller for both of the two monocular deprivation conditions (for eye-closed patching, F(2,22) = 17.213, p<0.001, partial $\eta^2$=0.610; for eye-open patching, F(2,22) = 24.371, p<0.001, partial $\eta^2$=0.689; one-way repeated-measures within-subjects analysis of variance [ANOVA]), while the contrast sensitivity of the PE becomes larger or does not change (for eye-closed patching, F(2,22) = 0.447, p=0.645, partial $\eta^2$=0.039; for eye-open patching, F(2,22) = 5.267, p=0.014, partial $\eta^2$=0.324; one-way repeated-measures within-subjects ANOVA).

We further calculated the contrast sensitivity ratio by dividing the contrast sensitivity of the PE into that of UPE for two patching conditions. The change of contrast sensitivity ratio after deprivation was shown in *Figure 4a*. If the UPE becomes weaker or PE becomes stronger, the contrast sensitivity ratio becomes more negative, otherwise, the ratio becomes more positive. We conducted a two-way repeated-measures ANOVA, with the patching conditions (two levels) and time points of measurements after deprivation (two levels) selected as within-subject factors. The results showed that there was significant difference between two time points (F(1,11) = 20.245, p<0.001, partial $\eta^2$=0.648), no difference between two patching conditions (F(1,11) = 0.811, p=*0.387*, partial $\eta^2$=0.069), and significant interaction of time point and condition (F(1,11) = 9.271, p=0.011, partial $\eta^2$=0.457). Post hoc Bonferroni test showed that the contrast sensitivity ratio changes between the eye-open and eye-closed patching at 0' was significantly different (p=0.023, *Figure 4b*).

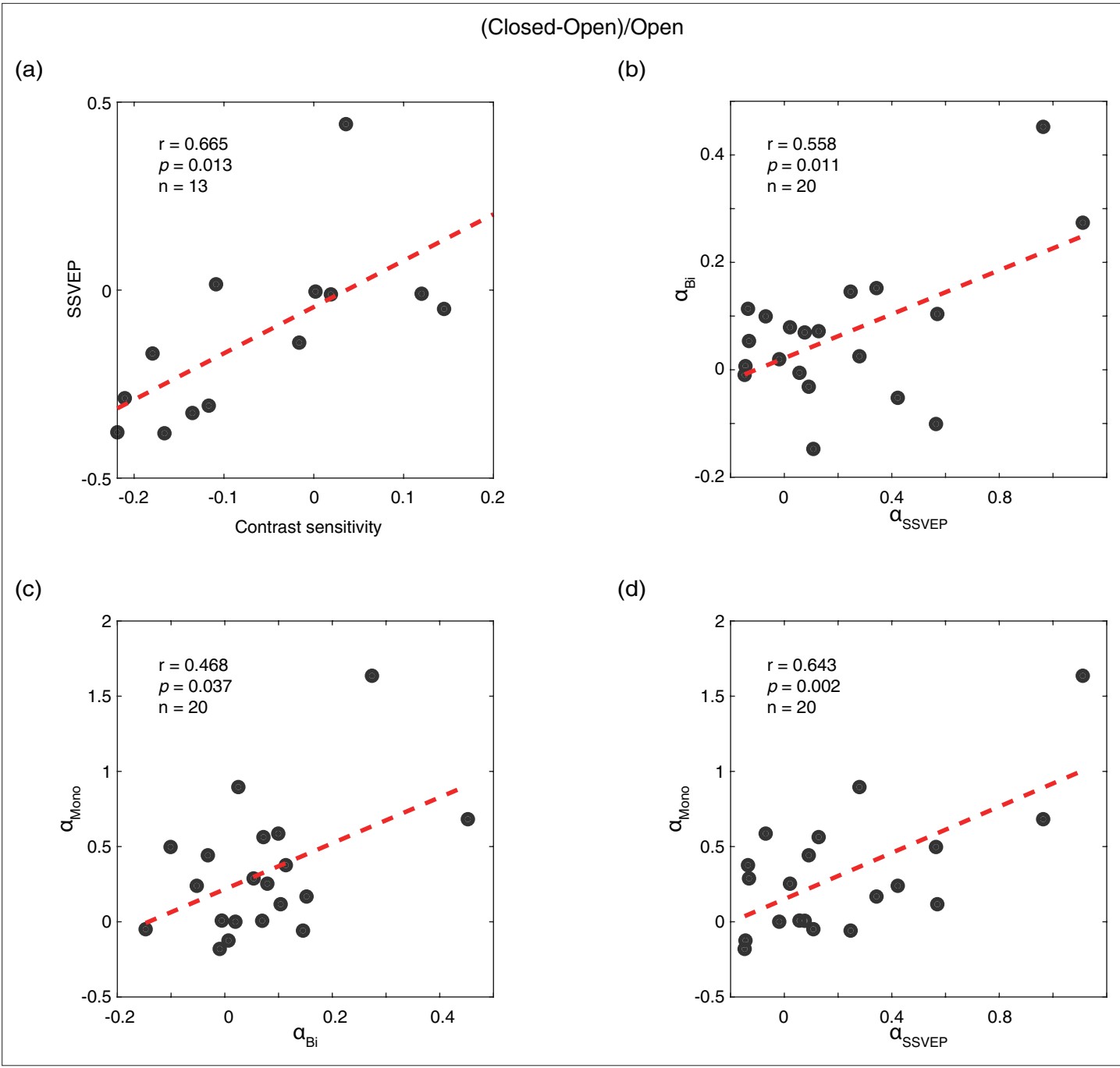

**Figure 2.** Correlations of the effects of eye open vs. eye close between different measurements. (**a**) Correlation between contrast sensitivity and SSVEP amplitude. (**b**) Correlation between peak alpha amplitude in the SSVEP sessions ($\alpha_{SSVEP}$) and the resting state when both eyes open/closed ($\alpha_{Bi}$). (**c**) Correlation between peak alpha amplitude in the resting state when both eyes open/closed ($\alpha_{Bi}$) and the resting state when PE open/closed ($\alpha_{mono}$). (**d**) Correlation between peak alpha amplitude in the SSVEP sessions ($\alpha_{SSVEP}$) and the resting state when PE open/closed ($\alpha_{mono}$). The difference of the eye-open and eye-closed conditions was calculated as: (closed-open)/open. Each dot represents one observer.

The online version of this article includes the following source data for figure 2:

**Source data 1.** Related to *Figure 2a, b, c and d*.

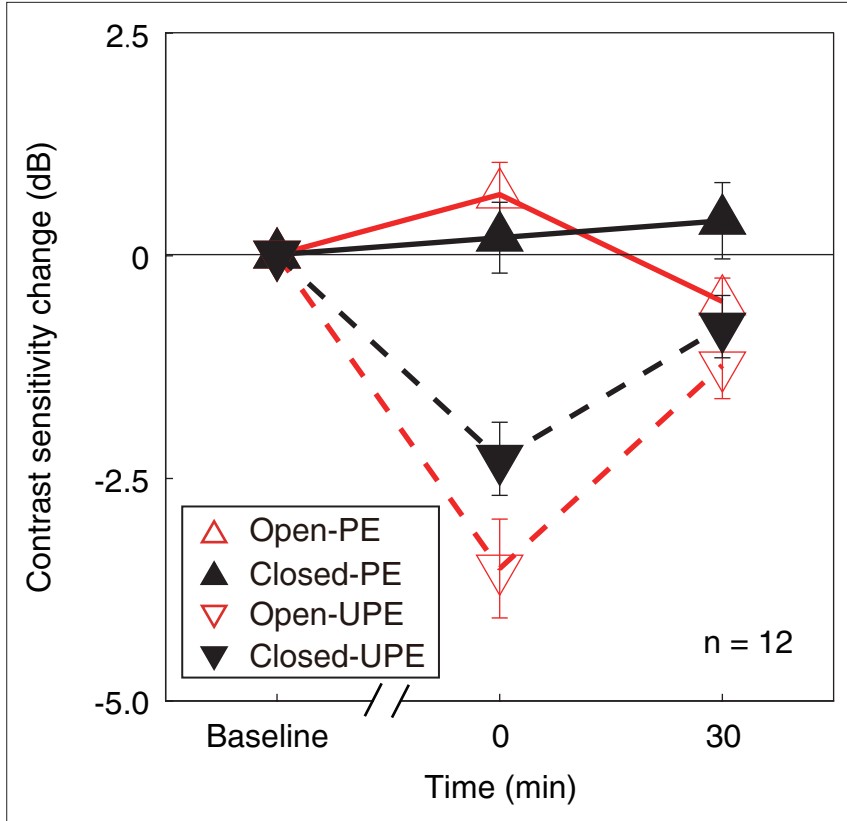

**Figure 3.** The change in monocular contrast sensitivity as a result of the monocular deprivation. The change in monocular contrast sensitivity as a result of the monocular deprivation was compared when the eye behind the patch remained open (the eye-open condition: red lines and open triangles) and when the eye behind the patch was kept closed (the eye-closed condition: black lines and filled triangles) for the PE (solid line with regular triangle) and UPE (dashed line with inverted triangle). The contrast sensitivity change in decibels (dB) was calculated as: $20 \times \log_{10}$(contrast sensitivity at post-measure session/contrast sensitivity at baseline). Error bars indicate one SEM across 12 subjects.

The online version of this article includes the following source data for figure 3:

**Source data 1.** Related to *Figure 3*.

## The aftereffects of 2.5-hr monocular patching on binocular combination

Since the results show that the contrast gain changes from monocular deprivation in the PE-closed condition is smaller than that in the PE-open condition, we wondered whether such an effect could also modulate the changes of sensory eye dominance after short-term monocular deprivation? We directly tested this by monitoring changes of sensory eye dominance as a result of monocular deprivation using a binocular phase combination task. After 2.5 hr of monocular patching in the PE-open condition, binocular perceived phase changed from –0.196±0.215° (mean ± SE) before patching to –14.196±3.184° immediately after removal of the patch. In contrast, in the PE-closed condition, binocular perceived phase changed from –0.214±0.308° to –6.357±1.962°. This means that the change in sensory eye dominance, as reflected by the binocular perceived phase, was 56.1% less in the PE-closed condition than that in the PE-open condition. The average change of perceived phase after patching where the patched (dominant) eye remains open (section B1) and where the patched (dominant) eye is kept closed (section B2) is plotted in *Figure 5a* as open red square symbols and filled black square symbols, respectively. The perceived phase changes in a more minus direction for both of the two monocular deprivation conditions. This means that after a 2.5 hr monocular patching, the contribution of PE to the binocularly fused percept becomes stronger. One-way repeated-measures within-subjects ANOVA showed that the binocular perceived phase significantly varied from baseline to post-measure sessions: for eye-closed patching, $F_{(3.022, 39.509)} = 7.126$, $p < 0.001$, partial $\eta^2 = 0.354$; for eye-open patching, $F_{(5, 65)} = 11.420$, $p < 0.001$, partial $\eta^2 = 0.468$. These results indicate that the PE, which was

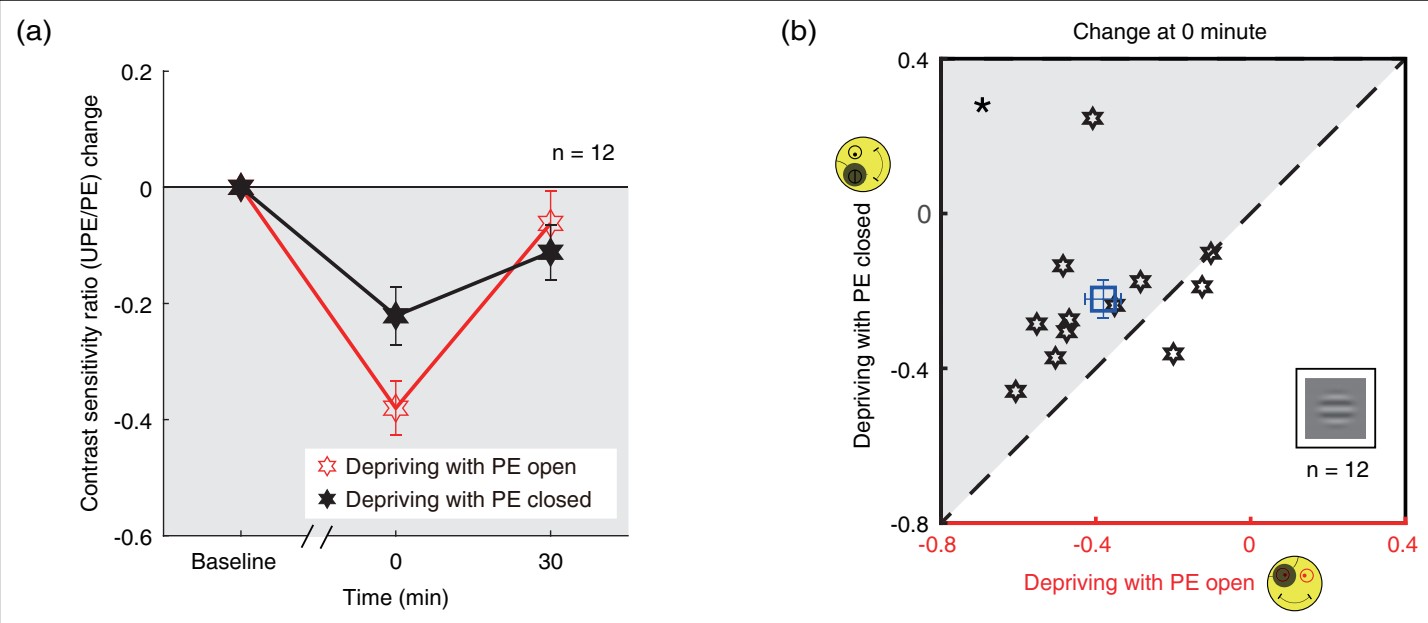

**Figure 4.** The change in contrast sensitivity ratio as a result of the monocular deprivation. (**a**) The change in contrast sensitivity ratio as a result of the monocular deprivation was compared when the eye behind the patch remained open (the eye-open condition: red lines and open hexagons) and when the eye behind the patch was kept closed (the eye-closed condition: black lines and filled hexagons). The contrast sensitivity ratio was calculated as: contrast sensitivity of UPE/contrast sensitivity of PE. The contrast sensitivity ratio change was calculated as: contrast sensitivity ratio at post-measure session − contrast sensitivity ratio at baseline. Error bars indicate one SEM across 12 subjects. (**b**) The average change of the post-measure session at 0' was compared for each subject for the two patching conditions. The open square symbol represents the averaged results. The dash line is the equality line. The gray area indicates where the eye-open patching produced more patching effect than the eye-closed patching. Error bars represent one SEM across 12 subjects. The black star indicates the significance of the two-tailed paired samples t-test, *: p<0.05.

The online version of this article includes the following source data for figure 4:

**Source data 1.** Related to *Figure 4a and b*.

the dominant eye, was significantly strengthened after both the eye-open patching (section B1) and the eye-closed patching (section B2).

One interesting result, which is also obvious in *Figure 5a*, is that the ocular dominance changes as a result of monocular deprivation are stronger in the eye-open condition than that in the eye-closed condition. A two-way repeated-measures within-subjects ANOVA also showed that the magnitude of the change of the perceived phase was significantly different between these two patching conditions (i.e., eye-open vs. eye-closed): $F_{(1,13)}$ = 10.265, p=0.007, partial $\eta^2$=0.441; the interaction between patching condition (i.e., eye-open vs. eye-closed) and the post-measure sessions (i.e., from 0' to 30') was not significant: $F_{(4,52)}$ = 1.553, p=0.201, partial $\eta^2$=0.107, indicating the different patching impacts between the eye-open and eye-closed patching was consistent within 30 min after the removal of the patch.

To further show the difference between these two patching conditions, we plotted individual averages of the perceived phase change of four post-measure sessions (0', 3', 6', 9') with eye-closed condition (section B2) as a function of that with eye-open condition (section B1) in *Figure 5b*. All subjects' data, except two, located above the equality line, indicating stronger patching effect in the eye-open condition than in the eye-closed condition. A two-tailed paired samples t-test also showed that there was a significant difference between these two conditions: $t_{(13)}$ = −3.276, p=0.006.

## The aftereffects of 2.5-hr monocular patching on binocular rivalry

Similar patching induced ocular dominance shifts were found using the binocular rivalry task (sections B3&B4, *Figure 6a*, red and black circles) showing a distinct increase in the dominance of the PE after either eye-open patching (section B3; *Figure 6a*, red circles) or eye-closed patching (section B4; *Figure 6a*, black circles). One-way repeated-measures within-subjects ANOVA also showed that the eye dominance ratio significantly varied from baseline to post-measure sessions: for eye-closed

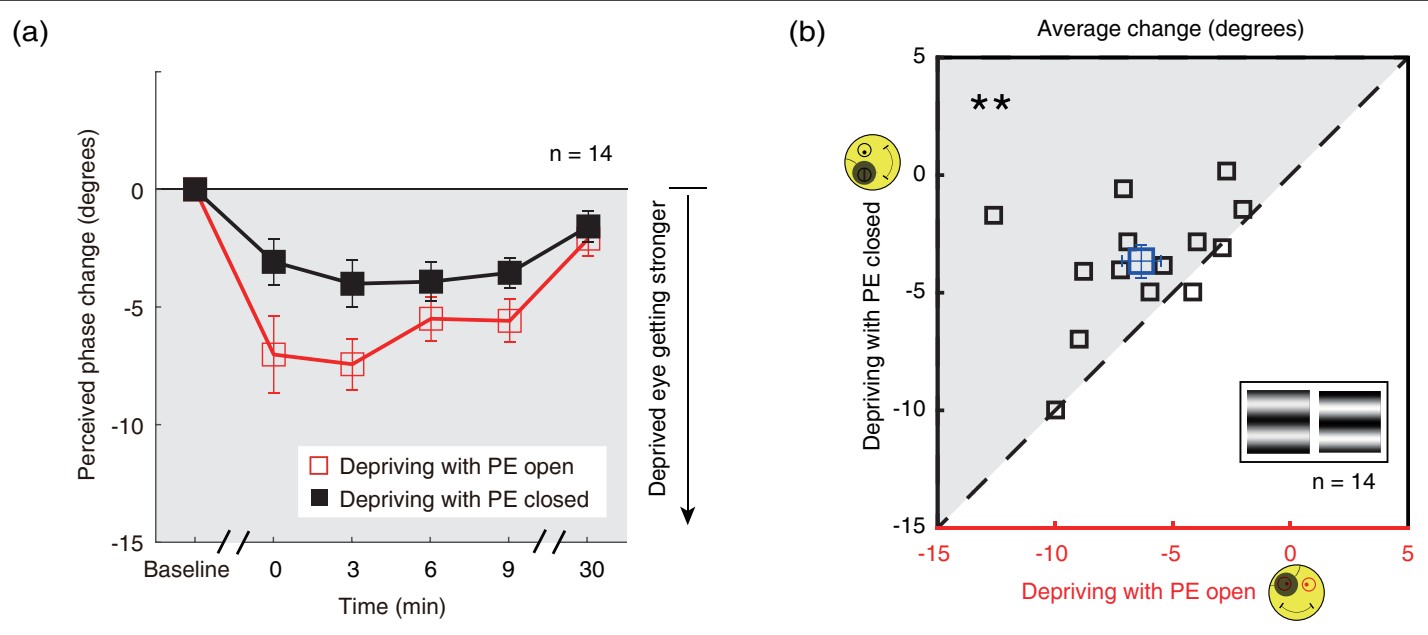

**Figure 5.** The change in ocular dominance as a result of the monocular deprivation in binocular combination. (**a**) The change in ocular dominance as a result of the monocular deprivation using phase combination task was compared for monocular patching where the eye behind the patch remained open (the eye-open condition – red lines and open squares) and when the eye behind the patch was closed (the eye-closed condition – black lines and filled squares). The perceived phase change was calculated as: perceived phase at post-measure session – perceived phase at baseline. Error bars indicate one SEM across 14 subjects. (**b**) The average change (degrees) of four post-measure sessions (0', 3', 6', 9') using the binocular phase combination task was compared for each subject for the two patching conditions. The open square symbol represents the averaged results. The dash line is the equality line. The gray area indicates where the eye-open patching produced more cumulated shift of ocular dominance than that of the eye-closed patching. Error bars represent one SEM across 14 subjects. The black star indicates the significance of the two-tailed paired samples t-test, **: p<0.001.

The online version of this article includes the following source data for figure 5:

**Source data 1.** Related to *Figure 5a and b*.

---

patching, F(5,65) = 17.047, p<0.001, partial $\eta^2$=0.567; for eye-open patching, F(5,65) = 34.987, p<0.001, partial $\eta^2$=0.729. After 2.5 hr of monocular patching in the PE-open condition, the eye dominance ratio changed from 0.946±0.035 (mean ± SE) before patching to 0.459±0.061 immediately after removal of the patch. In contrast, in the PE-closed condition, the eye dominance ratio changed from 0.922±0.042 to 0.508±0.075. This means that the change in sensory eye dominance, as reflected by the binocular rivalry, was 6.9% less in the PE-closed condition than that in the PE-open condition. The magnitude of the eye dominance ratio shift was also significantly different between the two patching conditions: F(1,13) = 5.256, p=0.039, partial $\eta^2$=0.288, without significant interaction between patching condition (i.e., eye-open vs. eye-closed) and the post-measure sessions (i.e., from 0' to 30'): F(4,52) = 2.009, p=0.107, partial $\eta^2$=0.134 (two-way repeated-measures ANOVA). In *Figure 6b*, the average of the perceived phase change of four post-measure sessions (0', 3', 6', 9') was compared for each subject for the eye-open and eye-closed patching conditions. All subjects, except three, showing stronger change in the eye-open condition than that in the eye-closed condition. A Wilcoxon signed-rank test also showed that there was a significant difference between two conditions: Z=–2.103, p=0.035.

## Discussion

In the absence of any visual stimulation, it is well accepted that the eyes open (REO) and eyes closed (REC) state, be it in the light or in the dark, can affect the resting state as reflected by changes in EEG spectral power and coherence in the Δ, θ, α1, α2, β1, β2, and γ frequency bands. Under complete darkness conditions, such changes cannot be related to exogenous visual stimulation, suggesting that the differences may be caused by the switching of involuntary preliminary attention from internally

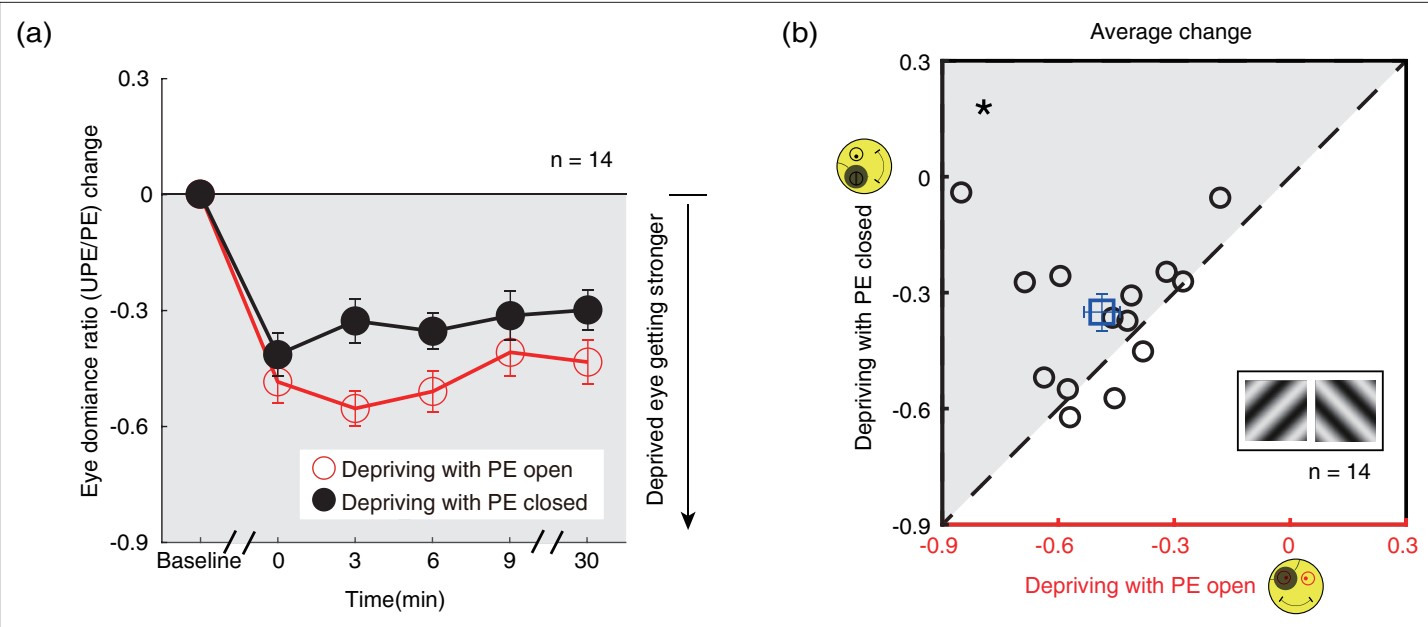

**Figure 6.** The change in ocular dominance as a result of the monocular deprivation in binocular rivalry. (**a**) The change in ocular dominance as a result of the monocular deprivation using binocular rivalry task was compared when the eye behind the patch remained open (the eye-open condition: red lines and open circles) and when the eye behind the patch was kept closed (the eye-closed condition: black lines and filled circles). The eye dominance ratio was calculated as: dominant duration of UPE/dominant duration of PE. The eye dominance ratio change was calculated as: eye dominance ratio at post-measure session − eye dominance ratio at baseline. Error bars indicate one SEM across 14 subjects. (**b**) The average change of four post-measure sessions (0′, 3′, 6′, 9′) using the binocular rivalry task was compared for each subject for the two patching conditions. The open square symbol represents the averaged results. The dash line is the equality line. The gray area indicates where the eye-open patching produced more cumulated shift of ocular dominance than the eye-closed patching. Error bars represent one SEM across 14 subjects. The black star indicates the significance of the Wilcoxon signed-rank test, *: p<0.05.

The online version of this article includes the following source data for figure 6:

**Source data 1.** Related to *Figure 6a and b*.

directed attention specific for the REC state to externally directed attention specific for the REO state (*Boytsova and Danko, 2010*). To our knowledge, we are the first to study the difference in brain state on visual sensitivity and changes in binocular balance in human adults. This was achieved by a monocular deprivation paradigm, with the PE that was being deprived of all visual input, being either open or closed. We show evidence that the PE-open and -closed states induce differences in alpha amplitude, which is quite similar with the differences in internal state when *both eyes* are open vs. when both eyes are closed.

The elevation of alpha oscillations when closing the PE, both for the resting state and SSVEP sessions, results in a reduction of contrast sensitivity of the UPE during deprivation of the other eye. This between-eye internal state effect suggests that these modulations of sensitivity by internal state manipulations are occurring at a binocular site. We also show that there is an endogenous modulation of the aftereffect from short-term monocular deprivation; the change in ocular dominance that results from such deprivation can be enhanced when the PE is kept open under the patch during the deprivation period. We show that this is true for binocular combination and also binocular rivalry even though the underlying mechanisms for these two tasks are thought to be very different; the former involves interocular gain control in early visual cortex (*Huang et al., 2010*) and the latter involves interocular competition and top-down influences from high-level visual areas (*Tong et al., 2006*).

The unique aspect of the present study is that we modulate the brain's internal state with one eye open and determine its impact on visual processing for both monocular contrast detection and binocular combination. By the manipulation of the internal states with one eye open or closed and assessing the visual sensitivity to the other eye, we can combine internal state manipulations with visual stimulation. We found that the alpha amplitude increased by 23.3% and 18.1% in the resting state and during visual stimulations, respectively, for the PE eye-closed condition compared to the PE eye-open

condition. We demonstrate that both visual sensitivity and the regulation of binocular balance can be modulated by internal neural state. We showed that by keeping the eye under the patch open, the immediate SSVEP amplitude and contrast sensitivity of UPE was increased, the resultant contrast gain and ocular dominance change when removing the occlude after 2.5 hr patching was enhanced. It should be emphasized that these changes are temporary because contrast gain and ocular dominance return to baseline levels after a certain period of time. However, there are differences in recovery time scales between tasks, which may be due to the fact that different tasks may reflect different aspects of striate and extrastriata function. In particular, the binocular perceived phase change in binocular combination which may rely on the involvement of phase-sensitive simple cells in the primary visual cortex (*Huang et al., 2010*) appears to recover to baseline within 30 min (*Figure 5a*). In contrast, changes in binocular rivalry, which involve more complex processing at multiple levels (involving extra-striate feedback as well as intrastriate processes) in the visual pathway (*Tong et al., 2006*), persist at the 30' post-measure session (*Figure 6a*). However, although we didn't measure further effects beyond 30 min, it would be premature to conclude that this manipulation would have limited clinical significance. This study involved the investigation of the effects of only one single session of 2.5 hr patching. Multiple daily patching sessions over a number of months are known have a more sustained effect and this sustained effect may be further enhanced by modulation of the internal state. In fact, studies have suggested that depriving the amblyopic eye for 2.5 hr can strengthen the amblyopic eye's contribution in binocular viewing (*Zhou et al., 2013b*), and repeated daily short-term monocular deprivation of the amblyopic eye not only can recover visual acuity of the amblyopic eye but lead to a more balanced binocular vision in adult amblyopes (*Lunghi et al., 2019c*; *Zhou et al., 2019*), which imply that monocular patching could have sustained therapeutic benefits to be implemented as a means to rebalance the visual system of amblyopic patients and improve their monocular acuity. Our study, combined with previous studies, suggest that it would be more effective to ensure that the eye remains open under the patch during treatment. On the other hand, the more traditional therapy for amblyopia is to patch the fellow good eye to force the amblyopic eye to improve. No one has ever considered whether the eye under the patch should be open or closed. The efficacy of this approach might be improved if the child was instructed to keep their fellow eye open under the patch. This may require a redesigned patch with enough eye clearance to ensure this is possible.

The differences that we found in brain states could be explained by the contrasting dynamics of GABA, which has implications in interpreting MRS measurements (*Kurcyus et al., 2018*). In complete darkness, GABA concentration decreases after eye opening (*Kurcyus et al., 2018*). GABA has been shown to be correlated with changes in sensory eye dominance following short-term monocular deprivation, where resting GABA concentration decreases after deprivation and this decrease in GABA correlates with the individuals' binocular changes (*Lunghi et al., 2015b*). Animal models have confirmed that GABA can mediate the neuroplastic change in primary visual cortex through a long-range cortical fibers that connect the large basket cells in the superficial cortical layers of the same and opposite ocular domains (*Buzás et al., 2001*; *Sengpiel et al., 1994*), and short-term monocular deprivation effects are associated with reduced GABAergic inhibition in layer 4 of V1 (*Reynaud et al., 2018*; *Tso et al., 2017*). Therefore, the most parsimonious explanation is that deprivation with the PE open is expected to reduce GABA levels more compared to deprivation with the PE closed, and thus induce more changes in sensory eye dominance as a result of short-term deprivation.

Our results suggest that having the eye open under the patch, even though this does not change the exogenous stimulation because the eye is occluded, it will result in an enhanced short-term effect for ocular dominance due solely to the internal neural state associated with a reduction of alpha inhibition.

## Materials and methods
### Participants
This study complied with the Declaration of Helsinki and was approved by the Institutional Review Boards of Wenzhou Medical University. The methods were carried out in accordance with the approved guidelines under the protocol 'Adult amblyopia: binocular visual deficits and rehabilitation' version #1 dated May 29, 2019. All subjects were naive to the purpose of the study, and provided written informed consent which included consent to process and preserve the data and

publish them in anonymous form. In total, 51 normal adults (age: 24.06±2.55 years of age; 22 males), with normal or corrected to normal vision (20/20 or better), participated in this study. In experiment 1, 20 subjects (age: 24.35±2.67 years of age; 12 males) participated in the EEG test, 18 subjects (age: 24.83±2.52 years of age; 10 males) participated in the behavioral test, and 13 subjects (age: 24.69±2.26 years of age; 7 males) participated in both EEG and behavioral tests. In experiment 2, 12 subjects (age: 24.08±1.04 years of age; 3 males) participated in the short-term patching study with monocular testing. In experiment 3, 14 subjects (age: 23.21±2.83 years of age; 4 males) participated in the short-term patching study with binocular testing. Observers wore their habitual optical correction if required. The sample sizes (n≥12 in all experiments) provide at least 80% power to detect a strong deprivation effect (Cohen's d>1) as suggested by previous studies of monocular deprivation using similar psychophysical tasks.

## Apparatus

In experiment 1, EEGs were recorded using a SynAmps amplifier system (Neuroscan) with a 64-channel cap (10–20 system). Both the EEG and behavioral test were conducted using a Windows computer. The programs were written with Matlab (Mathworks, Natick, MA, USA) and PsychToolBox 3.0 (*Brainard, 1997*). The stimulus was presented on a CRT monitor (NESOJXC FS210A) with GAMMA corrected, of which the resolution is 2048×1536 pixels, and the refresh rate is 60 Hz.

In experiment 2, the monocular contrast sensitivity measurement was conducted on an iMac computer using PsyKinematix software (*Beaudot, 2009*), and the stimuli were presented on a GAMMA corrected Built-In Retina LCD monitor (iMac, Apple, USA) in a dark room at a viewing distance of 60 cm. The monitor had a resolution of 2048×1152 pixels, a refresh rate of 60 Hz, and a comparable maximal luminance as goggles.

In experiment 3, sensory eye dominance measurements were conducted using a Mac computer running personally developed programs written with Matlab (Mathworks, Natick, MA, USA) and Psych-ToolBox 3.0 (*Brainard, 1997*). All stimuli were dichoptically presented using head mount goggles (Goovis pro, NED Optics, Shenzhen, China), which had a resolution of 1920×1080 pixels and a refresh rate of 60 Hz in each eye. The maximal luminance of the OLED goggles was 150 cd/m$^2$.

## Design and procedure
### EEG and behavioral tests on the immediate effects of open vs. close of the PE

EEGs were recorded in six conditions for each subject: (1) patch two eyes with both eyes open (2.5 min ×2), (2) patch two eyes with both eyes closed (2.5 min ×2), (3) patch one eye with both eyes open (5 min) and keep fixation on a gray background, (4) patch one eye with the PE closed (5 min) and keep fixation on a gray background, (5) patch one eye with both eyes open and present flickering sinewave plaids to the UPE (5 min), and (6) patch one eye with the PE closed and present the stimulus to the UPE (5 min). Both eyes open and closed conditions (conditions 1 and 2) were tested at both the beginning and the end of the EEG experiment, with an order of 'ABBA' or 'BAAB' that was balanced across subjects. The dominant eye (assessed by the hole-in-the-card test; *Dane and Dane, 2004*) was occluded in the monocular patching conditions by a black patch (conditions 3–6), which was specially designed by welding glasses (see *Figure 7—figure supplement 1*). All light would be blocked by this black patch. The only difference between the PE-open and -closed conditions is that we used a mild medical pressure-sensitive adhesive tape to ensure that observers' PE was closed. To prevent light leakage around, we also blocked the edge of the glasses with a shade cloth to ensure that no light would scatter around the patch into the PE. We also confirmed that the subjects could indeed not see any light when the UPE was closed before the test. In conditions 3 and 4, subjects were asked to keep fixation at the center of the screen on which we presented a mean luminance background of 68 cd/m$^2$ (i.e., the resting state) for 5 min. In conditions 5 and 6, a counterphase flickering sinewave plaid (2 cycles/°, 8° in diameter, 24% contrast, 7.5 Hz) was presented to the UPE to measure visually evoked signals (i.e., SSVEP). The stimulus was presented for 10 s for each trial, and 12 trials were collected for each condition. Subjects rested for 10 min in-between the PE-open and -closed conditions. The order of the PE-open and -closed conditions was counterbalanced across subjects.

EEGs were recorded at a sampling rate of 1000 Hz from the occipital and parietal electrodes, including all P, PO, O, and CB channels. Electrode impedances were kept below 10 kΩ. The electrode

REF on the cap between Cz and CPz was used as reference, and the electrode AFz was used as ground. EEG data were band-pass filtered from 1 to 30 Hz. For the resting-state conditions (conditions 1–4), EEG recordings of the first 10 s were removed to reduce the influence of artifacts, the remaining data were divided into 7 s epochs (i.e., same with the SSVEP-test time); for SSVEP recordings in conditions 5 and 6, EEGs of the first 2 s and the last 1 s were removed for each trial. To get the peak amplitude of alpha-band frequency, the amplitude spectrum for each epoch of each channel (occipital and parietal electrodes) was derived through fast Fourier transform (FFT) and was then averaged across epochs and channels. Then, Savitzky-Golay filter was performed on the averaged spectrum, and the peak value between 8 and 13 Hz was taken as the peak amplitude of alpha-band oscillation. To get the amplitude of SSVEP signals, we first averaged the EEG data across all trials and channels, and then derived the amplitude spectrum with FFT. The amplitude of the second harmonic at 15 Hz was taken as the SSVEP amplitude. To reduce the influence of large variations of SSVEP amplitude across subjects on the statistical results, the amplitude of SSVEPs of all conditions for each subject were normalized (divided) by the mean SSVEP amplitude across all conditions. The normalized amplitudes were then multiplied by the mean amplitudes across all subjects. Alpha amplitudes were normalized in the same way.

In a behavioral test, we measured contrast sensitivity for the UPE when the dominant eye was occluded. Eighteen observers, 13 of whom were the same as the EEG test, participated in the behavioral measurement. The behavioral test contains two conditions: the PE-open condition and the PE-close condition. The order of two conditions was counterbalanced across subjects. In this test, the contrast sensitivity was measured using an adjustment method. In particular, a 45° or 135° oriented sinewave grating (2 cycles/°, 8° in diameter) was presented to the UPE with an initial contrast lower than the observer's threshold. During the measurement, subjects were instructed to adjust the contrast of the grating by key pressing until they were just able to discriminate the orientation of the grating, and this contrast level was taken as the contrast threshold. Trials that the subject made a correct judgment about the orientation of the grating were included for the contrast threshold calculation. Contrast sensitivity was defined as the inverse of contrast threshold. There were 40 trials for each condition. Subjects practiced 40–60 trials before the formal test.

## The aftereffect of short-term patching on monocular contrast sensitivity

Twelve new observers participated in four patching sections. As with the psychophysical binocular test, each patching section consisted of three consecutive stages: a pre-deprivation measurement of contrast sensitivity (baseline), a 2.5 hr monocular deprivation stage, and a post-deprivation measurement of contrast sensitivity. For each observer, the dominant eye (assessed by the hole-in-the-card test; *Dane and Dane, 2004*) was chosen for short-term monocular deprivation. Observers were free to do any visual work, except sleeping or exercising under patching. During the monocular deprivation stage, observers' dominant eye was patched by a black patch with the PE open in sections M1&M3 and with the PE closed in sections M2&M4 (a medical pressure-sensitive adhesive tape was also used to ensure observers' PE was closed). Given that the deprivation effect does not last for more than 1 day according to a previous study (*Min et al., 2019*), each condition was tested on a separate day in a random order so that the interval between each two session was at least 24 hr, to prevent the perceptual changes after monocular deprivation from being carried over to a subsequent session.

An orientation discrimination task (*Figure 7a*) was used to measure monocular contrast sensitivity of the PE (sections M1&M2) and UPE (sections M3&M4). In this task, the monocular contrast function was measured using a constant stimuli method. In particular, a vertical or horizontal oriented sinewave grating (0.46 cycle/°, 4.33°×4.33°) was presented to the PE (sections M1&M2) or UPE (sections M3&M4). During the measurement, the unmeasured eye was patched by an opaque patch. Subjects were instructed to answer whether the orientation of the grating was vertical or horizontal by key pressing. Probabilities of correct identification were measured at six contrast levels; each level contains 75 trials, lasting for 15 min. We fitted the psychometric functions using Quick functions with parametric maximum likelihood estimation (*Watson, 1979*). The parameter alpha of the Quick function represents the threshold corresponding to 81.6% accuracy, whose mean and variances were determined by 500 times' bootstrap simulation (*Efron and Tibshirani, 1994*). Contrast sensitivity was defined as the inverse of contrast threshold. Using this approach, observers' monocular contrast

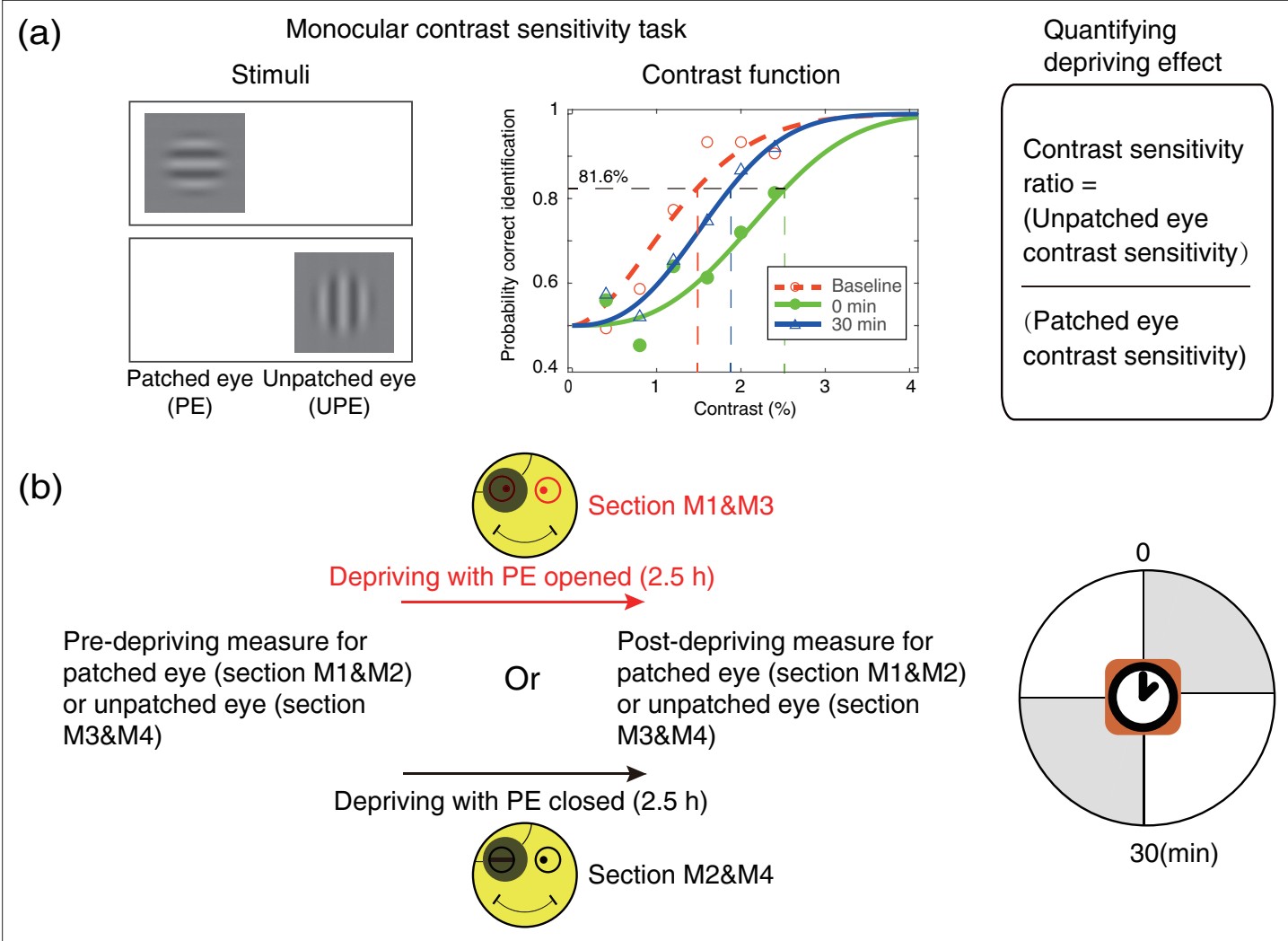

**Figure 7.** Experimental design and procedure of psychophysical monocular test. (**a**) The monocular contrast sensitivity task. The stimulus was a vertical or horizontal sinewave grating, which was presented to the patched eye or unpatched eye with the unmeasured eye occluded. Observers' contrast response functions were measured with constant stimuli method. Quick functions with maximum likelihood estimation were used to fit the contrast response function and derive the contrast thresholds. Contrast sensitivity was defined as the inverse of contrast threshold. Patching effect was quantified by the change of interocular contrast sensitivity ratio. (**b**) Monocular contrast sensitivity was measured before and after the 2.5 hr patching stage for patched eye or unpatched eye, started at multiple time points (0 and 30 min) after eye-open patching or eye-closed patching removal. (The black patch used for monocular deprivation is illustrated in *Figure 7—figure supplement 1*.)

The online version of this article includes the following figure supplement(s) for figure 7:

**Figure supplement 1.** The patch used to cover one eye.

sensitivity of each eye was assessed before the deprivation and at 0' and 30' after the completion of the 2.5 hr of monocular deprivation.

### The aftereffect of short-term patching on binocular imbalance

Fourteen observers participated in another four patching sections to measure the effects of monocular patching on binocular imbalance. Each patching section also consisted of three consecutive stages: a pre-deprivation measurement of sensory eye balance (baseline), a 2.5 hr monocular deprivation stage and a post-deprivation measurement of sensory eye balance. For each observer, the dominant eye (assessed by the hole-in-the-card test; *Dane and Dane, 2004*) was chosen for short-term monocular deprivation. Observers were free to do any visual work, except sleeping or exercising under patching. During the monocular deprivation stage, observers' dominant eye was

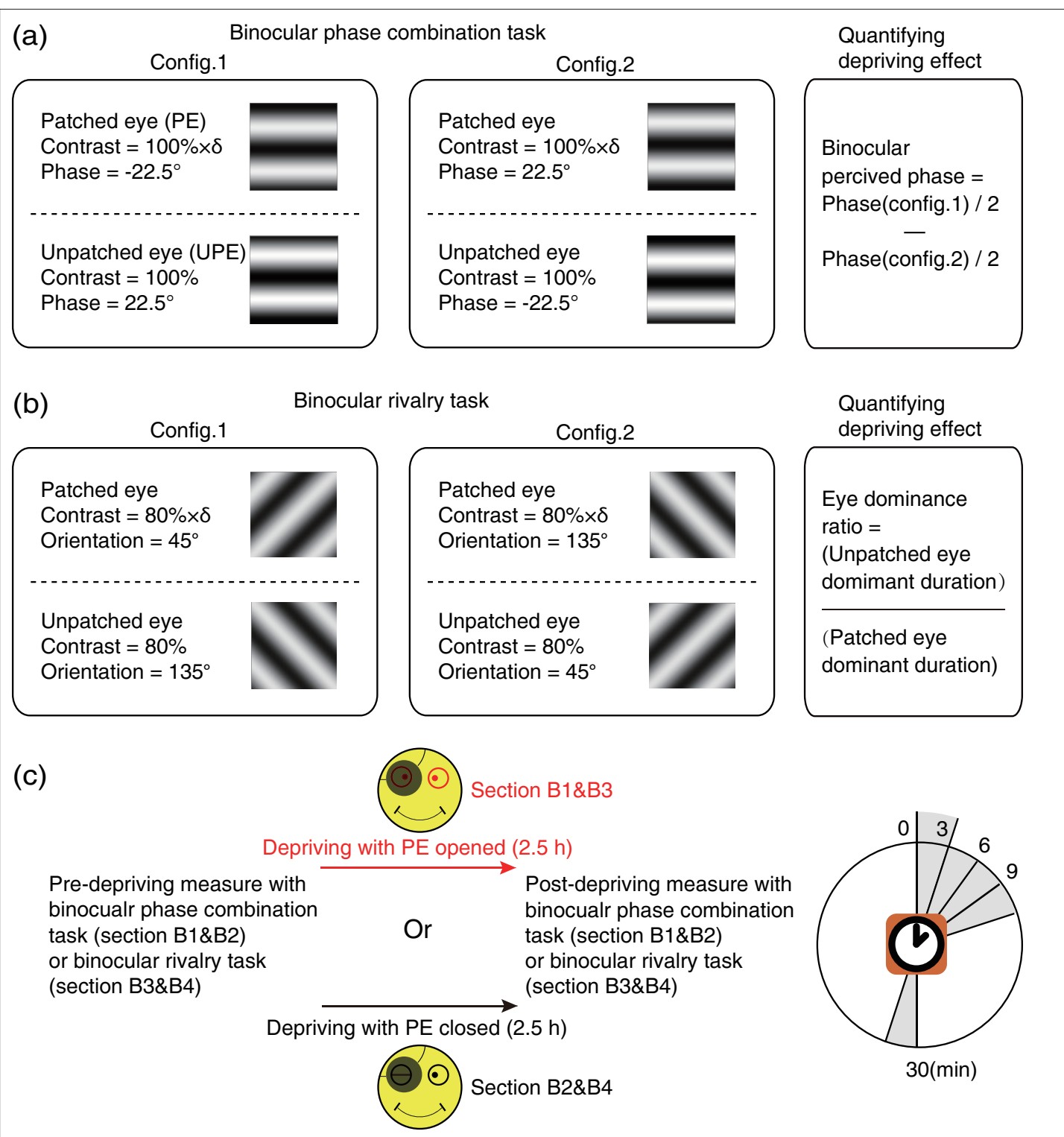

**Figure 8.** Experimental design and procedure of psychophysical binocular test. (**a**) The binocular phase combination task. The stimuli were two horizontal sinewave gratings with equal and opposite phase-shift of 22.5° relative to the horizontal center of the screen, which were dichoptically presented to the two eyes. Patching effect on sensory eye dominance was quantified by the change of binocularly perceived phase. (**b**) The binocular rivalry task. The stimuli were two orthogonal sinewave gratings, which were dichoptically presented to the two eyes. Patching effect on sensory eye dominance was quantified by the change of eye dominance ratio. (**c**) Sensory eye dominance was measured before and after the 2.5 hr patching stage by phase combination task or rivalry task, started at multiple time points (0–30 min) after eye-open patching or eye-closed patching removal. (The black patch used for monocular deprivation is illustrated in *Figure 7—figure supplement 1*.)

patched by a black patch with the PE open in sections B1&B3 and with the PE closed in sections B2&B4 (a medical pressure-sensitive adhesive tape was also used to ensure observers' PE was closed). Different deprivation conditions were conducted in a random order on different days for different observers.

In sections B1&B2, a binocular phase combination task (*Zhou et al., 2013a*) was used to quantitatively assess the degree of sensory eye balance, in which the binocularly perceived phase was measured and used as an index of sensory eye dominance (*Figure 8a*). In particular, two horizontal sinewave gratings (0.46 cycle/°, 4.33°×4.33°) with equal and opposite phase-shift of 22.5° relative to the center screen were dichoptically presented to the two eyes; the contrast of the UPE was fixed at 100% and the contrast of the PE was chosen close to an individuals' balance point in binocular phase combination before the deprivation (at that balance point, observer's two eyes are equally effective in binocular phase combination and the binocular perceptive phase is 0°). Two configurations were used to cancel any potential positional bias. In configuration 1, the phase-shift was +22.5° from horizontal in the UPE and –22.5° from horizontal in the PE and in configuration 2, the reverse. Each configuration was repeated eight times in one measurement session, in which 16 trials (8 repetitions × 2 configurations) were randomly interleaved. Normally, subjects could finish one measurement in 3 min after a short period of practice. Observers' binocular perceived phase was calculated by the averaged difference between the two configurations and was assessed before deprivation and at 0', 3', 6', 9', and 30' after the completion of the 2.5 hr of monocular deprivation. Thus, if the PE became stronger, the binocular perceived phase would be decreased, otherwise, if the PE became weaker, the binocular perceived phase would be increased.

In sections B3&B4, a binocular rivalry task was used to quantitatively assess the sensory eye balance. In this task, the interocular ratio of total phase duration (UPE/PE) was measured and used as an index of sensory eye dominance (*Figure 8b*). In particular, two vertically orthogonally oriented sinewave gratings (0.46 cycle/°, 4.33°×4.33°) were dichoptically presented to the two eyes; the contrast of the nondominant eye was fixed at 80%, and the contrast of the dominant eye was chosen close to an individuals' balance point in binocular rivalry before deprivation (at that balance point, observer's two eyes are equally effective in binocular rivalry and the ratio of total phase duration closed to 1). Each test session consists of two 90 s sub-blocks for two configurations: in configuration1, the orientation was 135° in the UPE and was 45° in the PE; in configuration 2, the orientation was 45° in the UPE and was 135° in the PE. Using this approach, observers' eye dominance ratio was calculated by the ratio between the UPE dominant duration and the PE dominant duration, and assessed before the deprivation and at 0', 3', 6', 9', and 30' after the completion of the 2.5 hr of monocular deprivation. Thus, if the PE became stronger, the sensory eye dominance ratio became more negative, otherwise, the ratio became more positive.

## Statistical analysis

SPSS v.23.0 (IBM Corporation, Armonk, NY, USA) was used for statistical analyses. We compared the immediate effects on EEG and behaviors between the PE-open and -closed condition using a two-tailed paired samples t-test with Holm-Bonferroni correction (*Holm, 1979*). We also conducted a Pearson correlation test to find the relationship between the effects on EEG and behaviors. Repeated-measures within-subjects ANOVA was applied to evaluate the effect of time after deprivation and the patching conditions (i.e., PE open and PE closed) on contrast sensitivity and binocular balance.

## Acknowledgements

This work was supported by the National Natural Science Foundation of China Grant (NSFC 31970975, 31871107, 31930053), the Natural Science Foundation for Distinguished Young Scholars of Zhejiang Province, China (LR22H120001), the Project of State Key Laboratory of Ophthalmology, Optometry and Vision Science, Wenzhou Medical University (No. J02-20210203, K03-20220102), the Canadian Institutes of Health Research Grants CCI-125686, NSERC grant 228103, and an ERA-NET Neuron grant (JTC2015), and the National Science and Technology Innovation 2030 Major Program (2022ZD0211900, 2021ZD0204200). The sponsor or funding organization had no role in the design or conduct of this research.

# Additional information

## Funding

| Funder | Grant reference number | Author |
| --- | --- | --- |
| National Natural Science Foundation of China | 31970975 | Jiawei Zhou |
| Natural Science Foundation for Distinguished Young Scholars of Zhejiang Province | LR22H120001 | Jiawei Zhou |
| Project of State Key Laboratory of Ophthalmology, Optometry and Vision Science, Wenzhou Medical University | J02-20210203 | Jiawei Zhou |
| Canadian Institutes of Health Research | CCI-125686 | Robert F Hess |
| Natural Sciences and Engineering Research Council of Canada | 228103 | Robert F Hess |
| ERA-NET Neuron | JTC2015 | Robert F Hess |
| National Natural Science Foundation of China | 31871107 | Peng Zhang |
| National Natural Science Foundation of China | 31930053 | Peng Zhang |
| Project of State Key Laboratory of Ophthalmology, Optometry and Vision Science, Wenzhou Medical University | K03-20220102 | Peng Zhang |
| National Science and Technology Major Project | 2022ZD0211900 | Peng Zhang |
| National Science and Technology Major Project | 2021ZD0204200 | Peng Zhang |

The funders had no role in study design, data collection and interpretation, or the decision to submit the work for publication.

## Author contributions

Yiya Chen, Yige Gao, Zhifen He, Data curation, Formal analysis, Writing – original draft, Writing – review and editing; Zhouyuan Sun, Yu Mao, Data curation; Robert F Hess, Conceptualization, Funding acquisition, Writing – review and editing; Peng Zhang, Conceptualization, Methodology, Writing – original draft, Writing – review and editing; Jiawei Zhou, Conceptualization, Supervision, Funding acquisition, Methodology, Writing – original draft, Writing – review and editing

## Author ORCIDs

Yiya Chen ⬤ http://orcid.org/0000-0002-8611-3158
Peng Zhang ⬤ http://orcid.org/0000-0002-9603-8454
Jiawei Zhou ⬤ http://orcid.org/0000-0003-4220-344X

## Ethics

This study complied with the Declaration of Helsinki and was approved by the Institutional Review Boards of Wenzhou Medical University. The methods were carried out in accordance with the approved guidelines under the protocol 'Adult amblyopia: binocular visual deficits and rehabilitation' version #1 dated 5/29/2019. All subjects were naive to the purpose of the study, and provided written informed

consent which included consent to process and preserve the data and publish them in anonymous form.

## Decision letter and Author response

Decision letter https://doi.org/10.7554/eLife.83815.sa1
Author response https://doi.org/10.7554/eLife.83815.sa2

## Additional files

### Supplementary files
• MDAR checklist

### Data availability
All data generated or analysed during this study are included in the manuscript and supporting files.

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
