## [Editor Report]

The authors report the results of three experiments assessing how one or both eyes open under a patch influence resting EEG activity, contrast sensitivity, and binocular balance in normally sighted subjects. Their results suggest that the state of eye-opening temporarily, but significantly, influences shifts in ocular dominance with relevance for the treatment of binocular visual disorders such as amblyopia that are treated with periodic monocular occlusion. The evidence supporting their conclusions is solid and the findings are important for the field.

---

## [Decision Letter]

**Decision letter after peer review:**

Thank you for submitting your article "The endogenous modulation of visual plasticity in human adults" for consideration by *eLife*. Your article has been reviewed by 2 peer reviewers, including Eric D Gaier as the Reviewing Editor and Reviewer #1, and the evaluation has been overseen by Lois Smith as the Senior Editor.

Essential revisions:

1) Revise language re: causation in manipulating α activity.

2) Revise language re: "neuroplasticity" and "endogenous."

3) Revise language in the Discussion re: GABA and pulvino-cortical inhibition (too speculative).

4) Improve the clarity of figures.

*Reviewer #1 (Recommendations for the authors):*

1. The statement from the abstract: "In this study we modulate the neural oscillations that determines the internal neural state to demonstrate that endogenous factors play a critical role in not only baseline contrast sensitivity but also the extent to which the adult visual system can undergo neuroplastic change in binocular balance" is representative of the major conclusions reached by the authors but is misleading in two important respects. First, while the investigators demonstrate changes to oscillatory activity as a function of eye open status under the patch, they do not directly manipulate oscillatory activity. It is acknowledged that the authors demonstrate a correlation between oscillatory effect and changes to SSVEP amplitude and contrast sensitivity, but again this does not demonstrate causality. Therefore, the effects of eye open status under the patch are only associated with these oscillatory changes, and their ability to conclude that the other observed effects result from changes in oscillatory activity is overstated. This should be modified in the abstract and throughout the text. Second, the term "neuroplastic change [plasticity]" can be generally applied to the subject of study here, but their use is misleading to audiences to whom these studies are most relevant. In particular, the ("neuroplastic") change imparted by these manipulations is essentially inconsequential, fleeting effects on sensory sensitivity with less than an hour. The term neuroplasticity is traditionally applied to long-term effects, especially when those effects are read out following cessation of experimental manipulations. This is not to undermine the potential importance of these findings, but they should be more accurately represented. The title and text should be modified to be more specific with regards to what was studied and what the main conclusion was rather than making such broad claims.

2. The abstract should focus less on background (currently 3 sentences) and more on the results (currently 1 sentence without specification of the direction of change). Additionally, the nature of the manipulations (see point #1 above) should be described.

3. The introduction defines what is meant by "endogenous" via contrast with what are defined as "exogenous" influences including image properties, exercise, visual pathology, and BMI. However, many would consider these to also engage "endogenous" factor as opposed to strictly exogenous factors such as pharmacologic, external brain stimulation, etc. Further to this point, the closed eye is mechanically taped shut per methods, and whether the patient's levator muscle is actually relaxed or active (as would be more of an "endogenous" factor as opposed to "exogenous" mechanical closure) may be important but is not controlled for here. As such, the term endogenous should be reconsidered and substituted.

4. The authors cite broad spectrum changes as a function of eye open/closed status, but they only seem to examine and show changes in α power. Additionally, though speculative, the discussion of GABAergic activity as a potential mechanism is appreciated.

5. The authors should report and comment on the magnitudes of these effects (EEG, CS, and binocular balance) and whether they are physiologically and/or clinically relevant changes. The normalization and extensive processing of data obscures this important perspective.

6. The application of these results to amblyopia is important, and the discussion paragraph lines 287-304 is much appreciated. However, this discussion (namely the suggestion to keep the patched eye open) is over-simplified; while contrast sensitivity may be enhanced for the amblyopic eye when the strong, fellow eye is kept open under the patch, the data clearly indicate that this benefit would be negated as soon as the patch is removed via increase contrast sensitivity and shift in binocular balance in favor of the stronger, patched eye for at least 30 minutes after a treatment session. A more carefully thought-out application of these data to the treatment of amblyopia would be appreciated for the suggestion on lines 299-304. The authors should consider expanding the discussion of the papers cited in this paragraph to inform readers of what the major findings were in those studies.

*Reviewer #2 (Recommendations for the authors):*

It is unclear how the use of the word "neuroplastic" differs from adaptation. Is there a difference?

Figure 1c- SSVEP amp increased with PE open vs closed- is there any possibility that light scatter around the patch could be entering the PE open?

Figure three y axis- contrast sensitivity change (in % change)?

Why plot figure 3 as change from baseline and then plot figure 4 as a ratio- both figures are normalized in different ways but neither show the actual data (i.e. contrast sensitivity values). Is the ratio calculated by dividing contrast sensitivity of PE /UPE or is it calculated by dividing CHANGE in contrast sensitivity of PE/UPE?

"Clearly, the perceived phase changes in a more minus direction for both of the two monocular deprivation conditions." This is not clear- what does more minus direction mean in terms of binocular balance.

"Compared with deprivation with PE closed, deprivation with PE open would be expected to reduce the GABA levels and thus induced more change in sensory eye dominance as a result of short-term deprivation. " Other than just presuming a reduction in GABA, it is unclear how the reduced GABA levels would induce a larger change in sensory eye dominance.

"On the other hand, previous studies show that occipital α inhibition originates from the pulvinar of the visual thalamus (Liu, de Zwart et al., 2012), our findings may suggest the role of pulvino-cortical inhibition in regulating ocular dominance plasticity in the early visual cortex." This claim is not well supported by the data.

Although the authors comment that the mechanism between binocular rivalry and binocular imbalance is different, they do not discuss any reasoning for why the phase change in binocular combination appears to almost recover to baseline, while the changes in binocular rivalry persist. Do the authors have any explanation for this difference?

What type of patch was used for the initial set of experiments? Was it the same type of patch for the monocular deprivation experiments (reported as a black patch)? Was there any perception of luminance changes through the patch? Was the patch positioned any differently for eyes open vs eyes closed conditions?

For the method of adjustment contrast sensitivity test, did you only include trials that the subject made a correct judgment about the orientation of the grating?

The legend of Figure 7 "started at multiple time points (0-30 min)" is confusing. My understanding is that CS was tested immediately after monocular deprivation (0 min) and then 30 minutes after deprivation.

The methods section could benefit from additional clarification regarding the timing of binocular balance and rivalry measures for Sections B1-B4. It says this was randomized, but there may be differences in monocular deprivation effects depending on if binocular phase combination or binocular rivalry was measured first or second. Did the authors compare these two groups to see if there was any pattern?

---

## [Author Response]

Essential revisions:1) Revise language re: causation in manipulating α activity.

Based on Reviewer #1’s suggestions, we have revised our demonstration in manipulating α activity in Title, Abstract and the main text, details of which can be found in our response to Reviewer #1’s Comment 1.

2) Revise language re: "neuroplasticity" and "endogenous."

In response to Reviewer #1 and Reviewer #2’s suggestion that the use of "neuroplastic change [plasticity]" may be misleading to audiences, we have rephrased the Title and emphasized the short-term changes in Introduction, and modified “endogenous factors” to “the internal neural states”. Details can be found in our responses to Reviewer #1’s Comments 1 and 3 and Reviewer #2’s Comment 1.

3) Revise language in the Discussion re: GABA and pulvino-cortical inhibition (too speculative).

Following the suggestions of Reviewer #1 and Reviewer #2 on the discussion of GABAergic activity, we have now provided more discussion of the link between GABA and changes in sensory eye dominance. Because the claim of pulvino-cortical inhibition is not well supported by the data and is not critical in the present study, we have removed it from the Discussion. Please see our responses to Reviewer #1’s Comment 4 and Reviewer #2’s Comments 6 and 7.

4) Improve the clarity of figures.

In response to Reviewer #1’s and Reviewer #2’s suggestions regarding the clarity of figures, we have now added units and highlighted in the figure legends how the data were normalized.

Reviewer #1 (Recommendations for the authors):1. The statement from the abstract: "In this study we modulate the neural oscillations that determines the internal neural state to demonstrate that endogenous factors play a critical role in not only baseline contrast sensitivity but also the extent to which the adult visual system can undergo neuroplastic change in binocular balance" is representative of the major conclusions reached by the authors but is misleading in two important respects. First, while the investigators demonstrate changes to oscillatory activity as a function of eye open status under the patch, they do not directly manipulate oscillatory activity. It is acknowledged that the authors demonstrate a correlation between oscillatory effect and changes to SSVEP amplitude and contrast sensitivity, but again this does not demonstrate causality. Therefore, the effects of eye open status under the patch are only associated with these oscillatory changes, and their ability to conclude that the other observed effects result from changes in oscillatory activity is overstated. This should be modified in the abstract and throughout the text. Second, the term "neuroplastic change [plasticity]" can be generally applied to the subject of study here, but their use is misleading to audiences to whom these studies are most relevant. In particular, the ("neuroplastic") change imparted by these manipulations is essentially inconsequential, fleeting effects on sensory sensitivity with less than an hour. The term neuroplasticity is traditionally applied to long-term effects, especially when those effects are read out following cessation of experimental manipulations. This is not to undermine the potential importance of these findings, but they should be more accurately represented. The title and text should be modified to be more specific with regards to what was studied and what the main conclusion was rather than making such broad claims.

Thanks for your comments. It’s right that we found an association between the effects of eye open status under the patch and oscillatory changes, rather than a causality. We have now revised our demonstration in Title, Abstract and the main text.

We agree with the comment that the use of "neuroplastic change [plasticity]" may be misleading to audiences. The changes in our study were a short-term effect, lasting for 30-90 minutes. We have consequently rephrased the Title and emphasized it in Introduction.

2. The abstract should focus less on background (currently 3 sentences) and more on the results (currently 1 sentence without specification of the direction of change). Additionally, the nature of the manipulations (see point #1 above) should be described.

We have now modified the Abstract:

“The adult human visual system maintains the ability to be altered by sensory deprivation. What has not been considered is whether the internal neural states modulate visual sensitivity to short-term monocular deprivation. In this study we manipulated the internal neural state and reported changes in intrinsic neural oscillations with a patched eye open or closed. We investigated the influence of eye open/eye closure on the unpatched eye’s contrast sensitivity and ocular dominance (OD) shifts induced by short-term monocular deprivation. The results demonstrate that internal neural states influence not only baseline contrast sensitivity but also the extent to which the adult visual system can undergo changes in ocular dominance.”

3. The introduction defines what is meant by "endogenous" via contrast with what are defined as "exogenous" influences including image properties, exercise, visual pathology, and BMI. However, many would consider these to also engage "endogenous" factor as opposed to strictly exogenous factors such as pharmacologic, external brain stimulation, etc. Further to this point, the closed eye is mechanically taped shut per methods, and whether the patient's levator muscle is actually relaxed or active (as would be more of an "endogenous" factor as opposed to "exogenous" mechanical closure) may be important but is not controlled for here. As such, the term endogenous should be reconsidered and substituted.

We now have modified “endogenous factors” to “internal neural states”.

4. The authors cite broad spectrum changes as a function of eye open/closed status, but they only seem to examine and show changes in α power. Additionally, though speculative, the discussion of GABAergic activity as a potential mechanism is appreciated.

Although there could be broad spectrum changes with eye open or closed, the most robust change is in the α band which is the hallmark of internal state changes (Boytsova and Danko, 2010). As shown by Figure 1a, our data indeed showed robust signals and significant changes in the α band. There were also weak signals in the β band (about 20 Hz), but without significant changes with eye open vs. closed.

We now clarified the reason for the focused analysis of the α band oscillation in the Results section.

In Line 100-102:

“Here, we focused on the α oscillation because it is the strongest intrinsic neural oscillation in the brain, which is the hallmark of internal state changes (Boytsova and Danko, 2010).”

In Line 110-111:

“There were also weak signals in the β band (Figure 1a, 20 Hz), but without significant changes in the eye-open than eye-closed condition.”

We have now discussed more about the link between GABA and the change in sensory eye dominance in Discussion (Line 414 – 439):

“The differences that we found in brain states could be explained by the contrasting dynamics of GABA, which has implications in interpreting MRS measurements (Kurcyus, Annac et al., 2018). In complete darkness, GABA concentration decreases after eye opening (Kurcyus, Annac et al., 2018). GABA has been shown to be correlated with changes in sensory eye dominance following short-term monocular deprivation, where resting GABA concentration decreases after deprivation and this decrease in GABA correlates with the individuals’ binocular changes (Lunghi, Emir et al., 2015). Animal models have confirmed that GABA can mediate the neuroplastic change in primary visual cortex through a long range cortical fibers that connect the large basket cells in the superficial cortical layers of the same and opposite ocular domains (Buzas, Eysel et al., 2001, Sengpiel, Blakemore et al., 1994), and short-term monocular deprivation effects are associated with reduced GABAergic inhibition in layer 4 of V1(Reynaud, Roux et al., 2018, Tso, Miller et al., 2017). Therefore, the most parsimonious explanation is that deprivation with the PE open is expected to reduce GABA levels more compared to deprivation with the PE closed, and thus induced more changes in sensory eye dominance as a result of short-term deprivation.”

5. The authors should report and comment on the magnitudes of these effects (EEG, CS, and binocular balance) and whether they are physiologically and/or clinically relevant changes. The normalization and extensive processing of data obscures this important perspective.

We now have highlighted the algorithm of changing values in the Figures, explained the change of original values in the Results, and added their physiological/clinical significance in the Discussion. Details are listed below.

Results, Line 138 – 142:

“Compared to the PE condition, the contrast sensitivity and SSVEP amplitude of the UPE was immediately decreased by 7.5% and 12.3%, whereas the amplitudes of αSSVEP, αBi and αmono was immediately enhanced by 18.1%, 6.2%, and 23.3%, respectively, in the PE closed condition.”

Results, Line 162 – 171:

“We measured monocular contrast sensitivity as the inverse of contrast threshold. After 2.5 hours of monocular patching in the PE open condition, the contrast sensitivity of the UPE changed from 103.086 ± 6.961 (mean ± SE) before patching to 70.017 ± 6.236 immediately after removal of the patch. This means that the short-term monocular deprivation temporally decreased the contrast sensitivity of UPE by 32.1%. In contrast, the contrast sensitivity of UPE decreased from 90.324 ± 6.791 to 69.324 ± 4.685 after 2.5 hours of monocular patching in the PE closed condition. This implies that the aftereffect of 2.5 hours of monocular patching on monocular contrast sensitivity was 34.5% less in the PE closed condition than in the PE open condition.”

Results, Line 239 – 245:

“After 2.5 hours of monocular patching in the PE open condition, binocular perceived phase changed from -0.196 ± 0.215 degrees (mean ± SE) before patching to -14.196 ± 3.184 degrees immediately after removal of the patch. In contrast, in the PE closed condition, binocular perceived phase changed from -0.214 ± 0.308 degrees to -6.357 ± 1.962 degrees. This means that the change in sensory eye dominance, as reflected by the binocular perceived phase, was 56.1% less in the PE closed condition than in the PE open condition.”

Results, Line 302 – 307:

“After 2.5 hours of monocular patching in the PE open condition, the eye dominance ratio changed from 0.946 ± 0.035 (mean ± SE) before patching to 0.459 ± 0.061 immediately after removal of the patch. In contrast, in the PE closed condition, the eye dominance ratio changed from 0.922 ± 0.042 to 0.508 ± 0.075. This means that the change in sensory eye dominance, as reflected by the binocular rivalry, was 6.9% less in the PE closed condition than in the PE open condition.”

Discussion, Line 372 – 408:

“We found that the α amplitude increased by 23.3% and 18.1% in the resting state and during visual stimulations, respectively, for the PE eye closed condition compared to the PE eye open condition. We demonstrate that both visual sensitivity and the regulation of binocular balance can be modulated by internal neural state. We showed that by keeping the eye under the patch open, the immediate SSVEP amplitude and contrast sensitivity of UPE was increased, the resultant contrast gain and ocular dominance change when removing the occlude after 2.5-hour patching was enhanced. It should be emphasized that these changes are temporary because contrast gain and ocular dominance return to baseline levels after a certain period of time. However, there are differences in recovery time scales between tasks, which may be due to the fact that different tasks may reflect different aspects of striate and extrastriata function. In particular, the binocular perceived phase change in binocular combination which may rely on the involvement of phase-sensitive simple cells in the primary visual cortex (Huang, Zhou et al., 2010) appears to recover to baseline within 30 minutes (Figure 5a). In contrast, changes in binocular rivalry, which involve more complex processing at multiple levels (involving extrastriate feedback as well as intrastriate processes) in the visual pathway (Tong, Meng et al., 2006), persist at the 30’ post-measure session (Figure 6a). However, although we didn’t measure further effects beyond 30 minutes, it would be premature to conclude that this manipulation would have limited clinical significance. This study involved the investigation of the effects of only one single session of 2.5-hour patching. Multiple daily patching sessions over a number of months are known have a more sustained effect and this sustained effect may be further enhanced by modulation of the internal state. In fact, studies have suggested that depriving the amblyopic eye for 2.5 hours can strengthen the amblyopic eye’s contribution in binocular viewing (Zhou, Thompson et al., 2013), and repeated daily short-term monocular deprivation of the amblyopic eye not only can recover visual acuity of the amblyopic eye but lead to a more balanced binocular vision in adult amblyopes (Lunghi, Sframeli et al., 2019, Zhou, He et al., 2019), which imply that monocular patching could have sustained therapeutic benefits to be implemented as a means to rebalance the visual system of amblyopic patients and improve their monocular acuity. Our study, combined with previous studies, suggest that it would be more effective to ensure that the eye remains open under the patch during treatment.”

6. The application of these results to amblyopia is important, and the discussion paragraph lines 287-304 is much appreciated. However, this discussion (namely the suggestion to keep the patched eye open) is over-simplified; while contrast sensitivity may be enhanced for the amblyopic eye when the strong, fellow eye is kept open under the patch, the data clearly indicate that this benefit would be negated as soon as the patch is removed via increase contrast sensitivity and shift in binocular balance in favor of the stronger, patched eye for at least 30 minutes after a treatment session. A more carefully thought-out application of these data to the treatment of amblyopia would be appreciated for the suggestion on lines 299-304. The authors should consider expanding the discussion of the papers cited in this paragraph to inform readers of what the major findings were in those studies.

We have now expanded the Discussion (Line 372 – 408):

“We found that the α amplitude increased by 23.3% and 18.1% in the resting state and during visual stimulations, respectively, for the PE eye closed condition compared to the PE eye open condition. We demonstrate that both visual sensitivity and the regulation of binocular balance can be modulated by internal neural state. We showed that by keeping the eye under the patch open, the immediate SSVEP amplitude and contrast sensitivity of UPE was increased, the resultant contrast gain and ocular dominance change when removing the occlude after 2.5-hour patching was enhanced. It should be emphasized that these changes are temporary because contrast gain and ocular dominance return to baseline levels after a certain period of time. However, there are differences in recovery time scales between tasks, which may be due to the fact that different tasks may reflect different aspects of striate and extrastriata function. In particular, the binocular perceived phase change in binocular combination which may rely on the involvement of phase-sensitive simple cells in the primary visual cortex (Huang, Zhou et al., 2010) appears to recover to baseline within 30 minutes (Figure 5a). In contrast, changes in binocular rivalry, which involve more complex processing at multiple levels (involving extrastriate feedback as well as intrastriate processes) in the visual pathway (Tong, Meng et al., 2006), persist at the 30’ post-measure session (Figure 6a). However, although we didn’t measure further effects beyond 30 minutes, it would be premature to conclude that this manipulation would have limited clinical significance. This study involved the investigation of the effects of only one single session of 2.5-hour patching. Multiple daily patching sessions over a number of months are known have a more sustained effect and this sustained effect may be further enhanced by modulation of the internal state. In fact, studies have suggested that depriving the amblyopic eye for 2.5 hours can strengthen the amblyopic eye’s contribution in binocular viewing (Zhou, Thompson et al., 2013), and repeated daily short-term monocular deprivation of the amblyopic eye not only can recover visual acuity of the amblyopic eye but lead to a more balanced binocular vision in adult amblyopes (Lunghi, Sframeli et al., 2019, Zhou, He et al., 2019), which imply that monocular patching could have sustained therapeutic benefits to be implemented as a means to rebalance the visual system of amblyopic patients and improve their monocular acuity. Our study, combined with previous studies, suggest that it would be more effective to ensure that the eye remains open under the patch during treatment.”

Reviewer #2 (Recommendations for the authors):It is unclear how the use of the word "neuroplastic" differs from adaptation. Is there a difference?

We agree that the boundary between adaptation and adult plasticity is not clear up to now. Also, Reviewer #1 suggested that the use of "neuroplastic change [plasticity]" may mislead the audiences because they always refer to long-term neuroplastic effect, while the change in our study were actually short-term changes that lasted only 30-90 minutes. Therefore, we have consequently rephrased the title and emphasized it in Introduction.

Figure 1c- SSVEP amp increased with PE open vs closed- is there any possibility that light scatter around the patch could be entering the PE open?

This is an important question. We control for that in all of our experiments. The coverage of one eye is achieved by a black patch that was specially designed by welding glasses (see Figure 7 —figure supplement 1). All light would be blocked by this black patch. The only difference between the PE open and PE closed conditions is that we used a medical pressure-sensitive adhesive tape to ensure that observers’ PE was closed. To prevent light leakage around, we also blocked the edge of the glasses with a shade cloth to ensure that no light would scatter around the patch into the PE. We also confirmed that the subjects could indeed not see any light when the unpatched eye was closed before the test. This is now elaborated in the main text in Lines 501 – 509.

Figure three y axis- contrast sensitivity change (in % change)?

We have now modified and calculated the contrast sensitivity change in decibels (dB), where dB = 20 × log_10_(contrast sensitivity at post-measure session/ contrast sensitivity at baseline). This is now clarified in the results and in the figure legend to Figure 3.

Why plot figure 3 as change from baseline and then plot figure 4 as a ratio- both figures are normalized in different ways but neither show the actual data (i.e. contrast sensitivity values). Is the ratio calculated by dividing contrast sensitivity of PE /UPE or is it calculated by dividing CHANGE in contrast sensitivity of PE/UPE?

Figure 3 shows the changes of contrast sensitivity for each eye. Figure 4 reflects an integrated effect between the two eyes. As shown in Figure 7a, the ratio is calculated from the contrast sensitivity of UPE/PE, and the amount of change is calculated as: contrast sensitivity ratio at post-measure session – contrast sensitivity ratio at baseline. This is now clarified in the figure legend to Figure 4. We have also provided the original contrast sensitivity values in Results, Line 162 – 171:

“We measured monocular contrast sensitivity as the inverse of contrast threshold. After 2.5 hours of monocular patching in the PE open condition, the contrast sensitivity of the UPE changed from 103.086 ± 6.961 (mean ± SE) before patching to 70.017 ± 6.236 immediately after removal of the patch. This means that the short-term monocular deprivation temporally decreased the contrast sensitivity of UPE by 32.1%. In contrast, the contrast sensitivity of UPE decreased from 90.324 ± 6.791 to 69.324 ± 4.685 after 2.5 hours of monocular patching in the PE closed condition. This implies that the aftereffect of 2.5 hours of monocular patching on monocular contrast sensitivity was 34.5% less in the PE closed condition than that in the PE open condition.”

"Clearly, the perceived phase changes in a more minus direction for both of the two monocular deprivation conditions." This is not clear- what does more minus direction mean in terms of binocular balance.

In Methods, we have clarified that if the PE became stronger, the binocular perceived phase would be decreased, otherwise, if the PE became weaker, the binocular perceived phase would be increased. Therefore, perceived phase changes in a more minus direction means PE becoming stronger (i.e. that is its contribution to binocular fusion is increased). We have now modified the sentence (Line 248 – 251):

“The perceived phase changes in a more minus direction for both of the two monocular deprivation conditions. This means that after a 2.5-hour monocular patching, the contribution of PE to the binocularly fused percept becomes stronger.”

"Compared with deprivation with PE closed, deprivation with PE open would be expected to reduce the GABA levels and thus induced more change in sensory eye dominance as a result of short-term deprivation. " Other than just presuming a reduction in GABA, it is unclear how the reduced GABA levels would induce a larger change in sensory eye dominance.

We have now discussed more about the link between GABA and the change in sensory eye dominance (Line 414 – 439):

“The differences that we found in brain states could be explained by the contrasting dynamics of GABA, which has implications in interpreting MRS measurements (Kurcyus, Annac et al., 2018). In complete darkness, GABA concentration decreases after eye opening (Kurcyus, Annac et al., 2018). GABA has been shown to be correlated with changes in sensory eye dominance following short-term monocular deprivation, where resting GABA concentration decreases after deprivation and this decrease in GABA correlates with the individuals’ binocular changes (Lunghi, Emir et al., 2015). Animal models have confirmed that GABA can mediate the neuroplastic change in primary visual cortex through a long range cortical fibers that connect the large basket cells in the superficial cortical layers of the same and opposite ocular domains (Buzas, Eysel et al., 2001, Sengpiel, Blakemore et al., 1994), and short-term monocular deprivation effects are associated with reduced GABAergic inhibition in layer 4 of V1(Reynaud, Roux et al., 2018, Tso, Miller et al., 2017). Therefore, the most parsimonious explanation is that deprivation with the PE open is expected to reduce GABA levels more compared to deprivation with the PE closed, and thus induced more changes in sensory eye dominance as a result of short-term deprivation.”

"On the other hand, previous studies show that occipital α inhibition originates from the pulvinar of the visual thalamus (Liu, de Zwart et al., 2012), our findings may suggest the role of pulvino-cortical inhibition in regulating ocular dominance plasticity in the early visual cortex." This claim is not well supported by the data.

Agree, and since it is not critical in the present study, we have now removed it from the Discussion.

Although the authors comment that the mechanism between binocular rivalry and binocular imbalance is different, they do not discuss any reasoning for why the phase change in binocular combination appears to almost recover to baseline, while the changes in binocular rivalry persist. Do the authors have any explanation for this difference?

Now clarified in Discussion (Line 377 – 396):

“We showed that by keeping the eye under the patch open, the immediate SSVEP amplitude and contrast sensitivity of UPE was increased, the resultant contrast gain and ocular dominance change when removing the occlude after 2.5-hour patching was enhanced. It should be emphasized that these changes are temporary because contrast gain and ocular dominance return to baseline levels after a certain period of time. However, there are differences in recovery time scales between tasks, which may be due to the fact that different tasks may reflect different aspects of striate and extrastriata function. In particular, the binocular perceived phase change in binocular combination which may rely on the involvement of phase-sensitive simple cells in the primary visual cortex (Huang, Zhou et al., 2010) appears to recover to baseline within 30 minutes (Figure 5a). In contrast, changes in binocular rivalry, which involve more complex processing at multiple levels (involving extrastriate feedback as well as intrastriate processes) in the visual pathway (Tong, Meng et al., 2006), persist at the 30’ post-measure session (Figure 6a). However, although we didn’t measure further effects beyond 30 minutes, it would be premature to conclude that this manipulation would have limited clinical significance. This study involved the investigation of the effects of only one single session of 2.5-hour patching. Multiple daily patching sessions over a number of months are known have a more sustained effect and this sustained effect may be further enhanced by modulation of the internal state.”

What type of patch was used for the initial set of experiments? Was it the same type of patch for the monocular deprivation experiments (reported as a black patch)? Was there any perception of luminance changes through the patch? Was the patch positioned any differently for eyes open vs eyes closed conditions?

The patches used for all experiments were the same, a black patch, which is now clarified in Methods for the initial set of experiments. The coverage of one eye is achieved by a black patch that was specially designed by welding glasses (see supplement Figure S1). All light would be blocked by this black patch. The only difference between the PE open and PE closed conditions is that we used a mild medical pressure-sensitive adhesive tape to ensure that observers’ PE was closed. To prevent light leakage around, we also blocked the edge of the glasses with a shade cloth to ensure that no light would scatter around the patch into the PE. We also confirmed that the subjects could indeed not see any light when the unpatched eye was closed before the test. This is now elaborated in the main text in Lines 481-489.

For the method of adjustment contrast sensitivity test, did you only include trials that the subject made a correct judgment about the orientation of the grating?

Yes. We have now clarified this detail in Methods (Line 549 – 550).

The legend of Figure 7 "started at multiple time points (0-30 min)" is confusing. My understanding is that CS was tested immediately after monocular deprivation (0 min) and then 30 minutes after deprivation.

Now changed it to “started at multiple time points (0 and 30 min)”.

The methods section could benefit from additional clarification regarding the timing of binocular balance and rivalry measures for Sections B1-B4. It says this was randomized, but there may be differences in monocular deprivation effects depending on if binocular phase combination or binocular rivalry was measured first or second. Did the authors compare these two groups to see if there was any pattern?

Given that the deprivation effect does not last for more than one day according to a previous study (Min et al., 2019), each condition was tested on a separate day so that the interval between each two session was at least 24 hours, to prevent the perceptual changes after monocular deprivation from being carried over to a subsequent session. The order was randomized for each subject. This is now elaborated in the main text in Lines 565-569.